# Goal-conditioned Imitation Learning

**Yiming Ding**[*]
Department of Computer Science
University of California, Berkeley
`dingyiming0427@berkeley.edu`

**Carlos Florensa**[*]
Department of Computer Science
University of California, Berkeley
`florensa@berkeley.edu`

**Mariano Phielipp**
Intel AI Labs
`mariano.j.phielipp@intel.com`

**Pieter Abbeel**
Department of Computer Science
University of California, Berkeley
`pabbeel@berkeley.edu`

## Abstract

Designing rewards for Reinforcement Learning (RL) is challenging because it needs to convey the desired task, be efficient to optimize, and be easy to compute. The latter is particularly problematic when applying RL to robotics, where detecting whether the desired configuration is reached might require considerable supervision and instrumentation. Furthermore, we are often interested in being able to reach a wide range of configurations, hence setting up a different reward every time might be unpractical. Methods like Hindsight Experience Replay (HER) have recently shown promise to learn policies able to reach many goals, without the need of a reward. Unfortunately, without tricks like resetting to points along the trajectory, HER might require many samples to discover how to reach certain areas of the state-space. In this work we propose a novel algorithm *goalGAIL*, which incorporates demonstrations to drastically speed up the convergence to a policy able to reach any goal, surpassing the performance of an agent trained with other Imitation Learning algorithms. Furthermore, we show our method can also be used when the available expert trajectories do not contain the actions or when the expert is suboptimal, which makes it applicable when only kinesthetic, third-person or noisy demonstrations are available. Our code is open-source [2].

## 1 Introduction

Reinforcement Learning (RL) has shown impressive results in a plethora of simulated tasks, ranging from attaining super-human performance in video-games [1, 2] and board-games [3], to learning complex locomotion behaviors [4, 5]. Nevertheless, these successes are shyly echoed in real world robotics [6, 7]. This is due to the difficulty of setting up the same learning environment that is enjoyed in simulation. One of the critical assumptions that are hard to obtain in the real world are the access to a reward function. Self-supervised methods have the power to overcome this limitation.

A very versatile and reusable form of self-supervision for robotics is to learn how to reach any previously observed state upon demand. This problem can be formulated as training a goal-conditioned policy [8, 9] that seeks to obtain the indicator reward of having the observation exactly match the goal. Such a reward does not require any additional instrumentation of the environment beyond the sensors the robot already has. But in practice, this reward is never observed because in continuous spaces like the ones in robotics, it is extremely rare to observe twice the exact same sensory input. Luckily, if we

---

[*]Equal contribution
[2]https://sites.google.com/view/goalconditioned-il/

are using an off-policy RL algorithm [10, 11], we can "relabel" a collected trajectory by replacing its goal by a state actually visited during that trajectory, therefore observing the indicator reward as often as we wish. This method was introduced as Hindsight Experience Replay [12] or HER, although it used special resets, and the reward was in fact an $\epsilon$-ball around the goal, which is only easy to interpret and use in lower-dimensional state-spaces. More recently the method was shown to work directly from vision with a special reward [13], and even only with the indicator reward of exactly matching observation and goal [14].

In theory these approaches could learn how to reach any goal, but the breadth-first nature of the algorithm makes that some areas of the space take a long time to be learned [15]. This is specially challenging when there are bottlenecks between different areas of the state-space, and random motion might not traverse them easily [16]. Some practical examples of this are pick-and-place, or navigating narrow corridors between rooms, as illustrated in Fig. 2 depicting the diverse set of environments we work with. In both cases a specific state needs to be reached (grasp the object, or enter the corridor) before a whole new area of the space is discovered (placing the object, or visiting the next room). This problem could be addressed by engineering a reward that guides the agent towards the bottlenecks, but this defeats the purpose of trying to learn without direct reward supervision. In this work we study how to leverage a few demonstrations that traverse those bottlenecks to boost the learning of goal-reaching policies.

Learning from Demonstrations, or Imitation Learning (IL), is a well-studied field in robotics [17, 18, 19]. In many cases it is easier to obtain a few demonstrations from an expert than to provide a good reward that describes the task. Most of the previous work on IL is centered around trajectory following, or doing a single task. Furthermore it is limited by the performance of the demonstrations, or relies on engineered rewards to improve upon them. In this work we first illustrate how IL methods can be extended to the goal-conditioned setting, and study a more powerful relabeling strategy that extracts additional information from the demonstrations. We then propose a novel algorithm, *goalGAIL*, and show it can outperform the demonstrator without the need of any additional reward. We also investigate how our method is more robust to sub-optimal experts. Finally, the method we develop is able to leverage demonstrations that do not include the expert actions. This considerably broadens its application in practical robotics, where demonstrations might be given by a motion planner, by kinesthetic demonstrations [20] (moving the agent externally, instead of using its own controls), or even by another agent [21]. To our knowledge, this is the first framework that can boost goal-conditioned policy learning with only state demonstrations.

## 2 Related Work

Imitation Learning is an alternative to reward crafting to train a desired behaviors. There are many ways to leverage demonstrations, from Behavioral Cloning [22] that directly maximizes the likelihood of the expert actions under the training agent policy, to Inverse Reinforcement Learning that extracts a reward function from those demonstrations and then trains a policy to maximize it [23, 24, 25]. Another formulation close to the latter is Generative Adversarial Imitation Learning (GAIL), introduced by Ho and Ermon [26]. GAIL is one of the building blocks of our own algorithm, and is explained in more details in the Preliminaries section.

Unfortunately most work in the field cannot outperform the expert, unless another reward is available during training [27, 28, 29], which might defeat the purpose of using demonstrations in the first place. Furthermore, most tasks tackled with these methods consist of tracking expert state trajectories [30, 31], but cannot adapt to unseen situations.

In this work we are interested in goal-conditioned tasks, where the objective is to reach any state upon demand [8, 9]. This kind of multi-task learning is pervasive in robotics [32, 33], but challenging and data-hungry if no reward-shaping is available. Relabeling methods like Hindsight Experience Replay [12] unlock the learning even in the sparse reward case [14]. Nevertheless, the inherent breath-first nature of the algorithm might still produce inefficient learning of complex policies. To overcome the exploration issue we investigate the effect of leveraging a few demonstrations. The closest prior work is by Nair et al. [34], where a Behavioral Cloning loss is used with a Q-filter. We found that a simple annealing of the Behavioral Cloning loss [35] works well and allows the agent to outperform demonstrator. Furthermore, we introduce a new relabeling technique of the expert trajectories that is

particularly useful when only few demonstrations are available. Finally we propose a novel algorithm *goalGAIL*, leveraging the recently shown compatibility of GAIL with off-policy algorithms.

## 3 Preliminaries

We define a discrete-time finite-horizon discounted Markov decision process (MDP) by a tuple $M = (\mathcal{S}, \mathcal{A}, \mathcal{P}, r, \rho_0, \gamma, H)$, where $\mathcal{S}$ is a state set, $\mathcal{A}$ is an action set, $\mathcal{P} : \mathcal{S} \times \mathcal{A} \times \mathcal{S} \to \mathbb{R}_+$ is a transition probability distribution, $\gamma \in [0, 1]$ is a discount factor, and $H$ is the horizon. Our objective is to find a stochastic policy $\pi_\theta$ that maximizes the expected discounted reward within the MDP, $\eta(\pi_\theta) = \mathbb{E}_\tau[\sum_{t=0}^T \gamma^t r(s_t, a_t, s_{t+1})]$. We denote by $\tau = (s_0, a_0, ..., )$ an entire state-action trajectory, where $s_0 \sim \rho_0(s_0)$, $a_t \sim \pi_\theta(\cdot|s_t)$, and $s_{t+1} \sim \mathcal{P}(\cdot|s_t, a_t)$. In the goal-conditioned setting that we use here, the policy and the reward are also conditioned on a "goal" $g \in \mathcal{S}$. The reward is $r(s_t, a_t, s_{t+1}, g) = \mathbb{1}[s_{t+1} == g]$, and hence the return is the $\gamma^h$, where $h$ is the number of time-steps to the goal. Given that the transition probability is not affected by the goal, $g$ can be "relabeled" in hindsight, so a transition $(s_t, a_t, s_{t+1}, g, r = 0)$ can be treated as $(s_t, a_t, s_{t+1}, g' = s_{t+1}, r = 1)$. Finally, we also assume access to $D$ trajectories $\{(s_0^j, a_0^j, s_1^j, ...)\}_{j=0}^D \sim \tau_{\text{expert}}$ that were collected by an expert attempting to reach the goals $\{g_j\}_{j=0}^D$ sampled uniformly among the feasible goals. Those trajectories must be approximately geodesics, meaning that the actions are taken such that the goal is reached as fast as possible.

In GAIL [26], a discriminator $D_\psi$ is trained to distinguish expert transitions $(s, a) \sim \tau_{\text{expert}}$, from agent transitions $(s, a) \sim \tau_{\text{agent}}$, while the agent is trained to "fool" the discriminator into thinking itself is the expert. Formally, the discriminator is trained to minimize $\mathcal{L}_{GAIL} = \mathbb{E}_{(s,a)\sim\tau_{\text{agent}}}[\log D_\psi(s, a)] + \mathbb{E}_{(s,a)\sim\tau_{\text{expert}}}[\log(1 - D_\psi(s, a))]$; while the agent is trained to maximize $\mathbb{E}_{(s,a)\sim\tau_{\text{agent}}}[\log D_\psi(s, a)]$ by using the output of the discriminator $\log D_\psi(s, a)$ as reward. Originally, the algorithms used to optimize the policy are on-policy methods like Trust Region Policy Optimization [36], but recently there has been a wake of works leveraging the efficiency of off-policy algorithms without loss in stability [37, 38, 39, 40]. This is a key capability that we exploit in our *goalGAIL* algorithm.

## 4 Demonstrations in Goal-conditioned tasks

In this section we describe methods to incorporate demonstrations into Hindsight Experience Replay [12] for training goal-conditioned policies. First we revisit adding a Behavioral Cloning loss to the policy update as in [34], then we propose a novel expert relabeling technique, and finally we formulate for the first time a goal-conditioned GAIL algorithm termed *goalGAIL*, and propose a method to train it with state-only demonstrations.

### 4.1 Goal-conditioned Behavioral Cloning

The most direct way to leverage demonstrations $\{(s_0^j, a_0^j, s_1^j, ...)\}_{j=0}^D$ is to construct a data-set $\mathcal{D}$ of all state-action-goal tuples $(s_t^j, a_t^j, g^j)$, and run supervised regression. In the goal-conditioned case and assuming a deterministic policy $\pi_\theta(s, g)$, the loss is:

$$\mathcal{L}_{\text{BC}}(\theta, \mathcal{D}) = \mathbb{E}_{(s_t^j, a_t^j, g^j)\sim\mathcal{D}}\left[\|\pi_\theta(s_t^j, g^j) - a_t^j\|_2^2\right] \tag{1}$$

This loss and its gradient are computed without any additional environments samples from the trained policy $\pi_\theta$. This makes it particularly convenient to combine a gradient descend step based on this loss together with other policy updates. In particular we can use a standard off-policy Reinforcement Learning algorithm like DDPG [10], where we fit the $Q_\phi(a, s, g)$, and then estimate the gradient of the expected return as:

$$\nabla_\theta \hat{J} = \frac{1}{N}\sum_{i=1}^N \nabla_a Q_\phi(a, s, g)\nabla_\theta \pi_\theta(s, g) \tag{2}$$

In our goal-conditioned case, the $Q$ fitting can also benefit from "relabeling" like done in HER [12]. The improvement guarantees with respect to the task reward are lost when we combine the BC and

the deterministic policy gradient updates, but this can be side-stepped by either applying a Q-filter $\mathbb{1}\{Q(s_t, a_t, g) > Q(s_t, \pi(s_t, g), g)\}$ to the BC loss as proposed in [34], or by annealing it as we do in our experiments, which allows the agent to eventually outperform the expert.

## 4.2 Relabeling the expert

The expert trajectories have been collected by asking the expert to reach a specific goal $g^j$. But they are also valid trajectories to reach any other state visited within the demonstration! This is the key motivating insight to propose a new type of relabeling: if we have the transitions $(s_t^j, a_t^j, s_{t+1}^j, g^j)$ in a demonstration, we can also consider the transition $(s_t^j, a_t^j, s_{t+1}^j, g' = s_{t+k}^j)$ as also coming from the expert! Indeed that demonstration also went through the state $s_{t+k}^j$, so if that was the goal, the expert would also have generated this transition. This can be understood as a type of data augmentation leveraging the assumption that the given demonstrations are geodesics (they are the faster way to go from any state in it to any future state in it). It will be particularly effective in the low data regime, where not many demonstrations are available. The effect of expert relabeling can be visualized in the four rooms environment as it's a 2D task where states and goals can be plotted. In Fig. 1 we compare the final performance of two agents, one trained with pure Behavioral Cloning, and the other one also using expert relabeling.

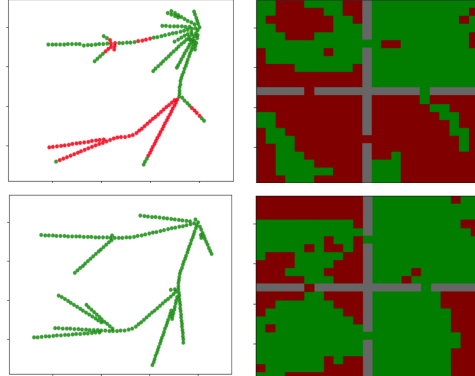

(a) Performance on reaching states visited in the 20 given demonstrations. The states are green if reached by the policy when attempting so, and red otherwise.

(b) Performance on reaching any possible state. Each cell is colored green if the policy can reach the center of it when attempting so, and red otherwise.

Figure 1: Policy performance on reaching different goals in the four rooms, when training with standard Behavioral Cloning (top row) or with our expert relabeling (bottom).

## 4.3 Goal-conditioned GAIL with Hindsight

The compounding error in Behavioral Cloning might make the policy deviate arbitrarily from the demonstrations, and it requires too many demonstrations when the state dimension increases. The first problem is less severe in our goal-conditioned case because in fact we do want to visit and be able to purposefully reach all states, even the ones that the expert did not visit. But the second drawback will become pressing when attempting to scale this method to practical robotics tasks where the observations might be high-dimensional sensory input like images. Both problems can be mitigated by using other Imitation Learning algorithms that can leverage additional rollouts collected by the learning agent in a self-supervised manner, like GAIL [26]. In this section we extend the formulation of GAIL to tackle goal-conditioned tasks, and then we detail how it can be combined with HER [12], which allows the agent to outperform the demonstrator and generalize to reaching all goals. We call the final algorithm *goalGAIL*.

We describe the key points of *goalGAIL* below. First of all, the discriminator needs to also be conditioned on the goal which results in $D_\psi(a, s, g)$, and be trained by minimizing

$$
\begin{aligned}
\mathcal{L}_{GAIL}(D_\psi, \mathcal{D}, \mathcal{R}) \; = \; & \mathbb{E}_{(s,a,g)\sim\mathcal{R}}[\log D_\psi(a, s, g)] \; + \\
& \mathbb{E}_{(s,a,g)\sim\mathcal{D}}[\log(1 - D_\psi(a, s, g))].
\end{aligned}
\tag{3}
$$

Once the discriminator is fitted, we can run our favorite RL algorithm on the reward $\log D_\psi(a_t^h, s_t^h, g^h)$. In our case we used the off-policy algorithm DDPG [10] to allow for the relabeling techniques outlined above. In the goal-conditioned case we intepolate between the GAIL reward described above and an indicator reward $r_t^h = \mathbb{1}[s_{t+1}^h == g^h]$. This combination is slightly tricky because now the fitted $Q_\phi$ does not have the same clear interpretation it has when only one of the two rewards is used [14] . Nevertheless, both rewards are pushing the policy towards the goals, so it shouldn't be too conflicting. Furthermore, to avoid any drop in final performance, the weight of the reward coming from GAIL $\delta_{GAIL}$ can be annealed. The final proposed algorithm *goalGAL*, together with the expert relabeling technique is formalized in Algorithm 1.

**Algorithm 1** Goal-conditioned GAIL with Hindsight: *goalGAIL*

---

1: **Input:** Demonstrations $\mathcal{D} = \left\{(s_0^j, a_0^j, s_1^j, ..., g^j)\right\}_{j=0}^{D}$, replay buffer $\mathcal{R} = \{\}$, policy $\pi_\theta(s, g)$, discount $\gamma$, hindsight probability $p$
2: **while** not done **do**
3:      *# Sample rollout*
4:      $g \sim \texttt{Uniform}(\mathcal{S})$
5:      $\mathcal{R} \leftarrow \mathcal{R} \cup (s_0, a_0, s_1, ...)$ sampled using $\pi(\cdot, g)$
6:      *# Sample from expert buffer and replay buffer*
7:      $\left\{(s_t^j, a_t^j, s_{t+1}^j, g^j)\right\} \sim \mathcal{D}, \left\{(s_t^i, a_t^i, s_{t+1}^i, g^i)\right\} \sim \mathcal{R}$
8:      *# Relabel agent transitions*
9:      **for** each $i$, with probability $p$ **do**
10:        $g^i \leftarrow s_{t+k}^i, \quad k \sim \text{Unif}\{t+1, \ldots, T^i\}$                 ▷ Use *future* HER strategy
11:      **end for**
12:      *# Relabel expert transitions*
13:      $g^j \leftarrow s_{t+k'}^j, \quad k' \sim \text{Unif}\{t+1, \ldots, T^j\}$
14:      $r_t^h = \mathbb{1}\left[s_{t+1}^h == g^h\right]$
15:      $\psi \leftarrow \min_\psi \mathcal{L}_{GAIL}(D_\psi, \mathcal{D}, \mathcal{R})$ (Eq. 3)
16:      $r_t^h = (1 - \delta_{GAIL})r_t^h + \delta_{GAIL} \log D_\psi(a_t^h, s_t^h, g^h)$        ▷ Add annealed GAIL reward
17:      *# Fit $Q_\phi$*
18:      $y_t^h = r_t^h + \gamma Q_\phi(\pi(s_{t+1}^h, g^h), s_{t+1}^h, g^h)$         ▷ Use target networks $Q_{\phi'}$ for stability
19:      $\phi \leftarrow \min_\phi \sum_h \|Q_\phi(a_t^h, s_t^h, g^h) - y_t^h\|$
20:      *# Update Policy*
21:      $\theta += \alpha \nabla_\theta \hat{J}$ (Eq. 2)
22:      Anneal $\delta_{GAIL}$                                             ▷ Ensures outperforming the expert
23: **end while**

---

## 4.4 Use of state-only Demonstrations

Both Behavioral Cloning and GAIL use state-action pairs from the expert. This limits the use of the methods, combined or not with HER, to setups where the exact same agent was actuated to reach different goals. Nevertheless, much more data could be cheaply available if the action was not required. For example, non-expert humans might not be able to operate a robot from torque instructions, but might be able to move the robot along the desired trajectory. This is called a kinesthetic demonstration. Another type of state-only demonstration could be the one used in third-person imitation [21], where the expert performed the task with an embodiment different from the agent that needs to learn the task. This has mostly been applied to the trajectory-following case. In our case every demonstration might have a different goal.

Furthermore, we would like to propose a method that not only leverages state-only demonstrations, but can also outperform the quality and coverage of the demonstrations given, or at least generalize to similar goals. The main insight we have here is that we can replace the action in the GAIL formulation by the next state $s'$, and in most environments this should be as informative as having access to the action directly. Intuitively, given a desired goal $g$, it should be possible to determine if a transition $s \rightarrow s'$ is taking the agent in the right direction. The loss function to train a discriminator able to tell apart the current agent and expert demonstrations (always transitioning towards the goal) is simply:

$$\mathcal{L}_{GAIL^s}(D_\psi^s, \mathcal{D}, \mathcal{R}) = \mathbb{E}_{(s,s',g)\sim\mathcal{R}}[\log D_\psi^s(s, s', g)] + \mathbb{E}_{(s,s',g)\sim\mathcal{D}}[\log(1 - D_\psi^s(s, s', g))].$$

## 5 Experiments

We are interested in answering the following questions:

- Without supervision from reward, can *goalGAIL* use demonstrations to accelerate the learning of goal-conditioned tasks and outperform the demonstrator?
- Is the *Expert Relabeling* an efficient way of doing data-augmentation on the demonstrations?
- Compared to Behavioral Cloning methods, is *goalGAIL* more robust to expert action noise?
- Can *goalGAIL* leverage state-only demonstrations equally well as full trajectories?

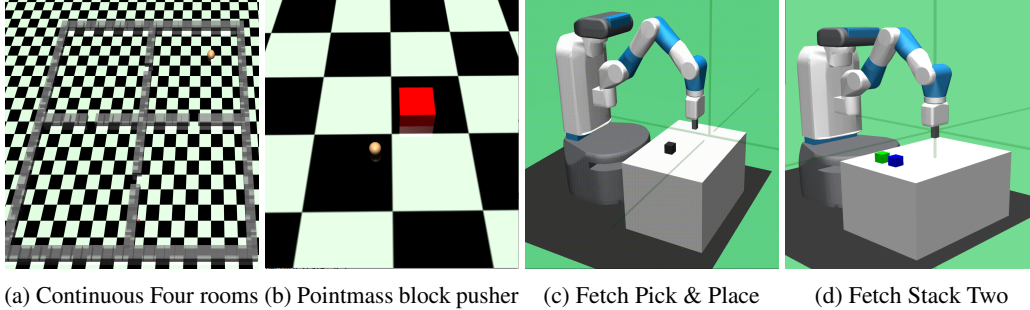

(a) Continuous Four rooms (b) Pointmass block pusher    (c) Fetch Pick & Place    (d) Fetch Stack Two

Figure 2: Four continuous goal-conditioned environments where we tested the effectiveness of the proposed algorithm *goalGAIL* and expert relabeling technique.

We evaluate these questions in four different simulated robotic goal-conditioned tasks that are detailed in the next subsection along with the performance metric used throughout the experiments section. All the results use 20 demonstrations reaching uniformly sampled goals. All curves have 5 random seeds and the shaded area is one standard deviation.

## 5.1  Tasks

Experiments are conducted in four continuous environments in MuJoCo [41]. The performance metric we use in all our experiments is the percentage of goals in the feasible goal space the agent is able to reach. We call this metric coverage. To estimate this percentage we sample feasible goals uniformly, and execute a rollout of the current policy. It is consider a success if the agent reaches within $\epsilon$ of the desired goal.

**Four rooms environment**: This is a continuous twist on a well studied problem in the Reinforcement Learning literature. A point mass is placed in an environment with four rooms connected through small openings as depicted in Fig. 2a. The action space of the agent is continuous and specifies the desired change in state space, and the goals-space exactly corresponds to the state-space.

**Pointmass Block Pusher**: In this task, a Pointmass needs to navigates itself to the block, push the block to a desired position $(x, y)$ and then eventually stops a potentially different spot $(a, b)$. The action space is two dimensional as in four rooms environment. The goal space is four dimensional and specifies $(x, y, a, b)$.

**Fetch Pick and Place**: This task is the same as the one described by Nair et al. [34], where a fetch robot needs to pick a block and place it in a desired point in space. The control is four-dimensional, corresponding to a change in $(x, y, z)$ position of the end-effector as well as a change in gripper opening. The goal space is three dimensional and is restricted to the position of the block.

**Fetch Stack Two**: A Fetch robot stacks two blocks on a desired position, as also done in Nair et al. [34]. The control is the same as in Fetch Pick and Place while the goal space is now the position of two blocks, which is six dimensional.

## 5.2  Goal-conditioned GAIL with Hindsight: goalGAIL

In goal-conditioned tasks, HER [12] should eventually converge to a policy able to reach any desired goal. Nevertheless, this might take a long time, specially in environments where there are bottlenecks that need to be traversed before accessing a whole new area of the goal space. In this section we show how the methods introduced in the previous section can leverage a few demonstrations to improve the convergence speed of HER. This was already studied for the case of Behavioral Cloning by [34], and in this work we show we also get a benefit when using GAIL as the Imitation Learning algorithm, which brings considerable advantages over Behavioral Cloning as shown in the next sections. In all four environments, we observe that our proposed method *goalGAIL* considerably outperforms the two baselines it builds upon: HER and GAIL. HER alone has a very slow convergence, although as expected it ends up reaching the same final performance if run long enough. On the other hand GAIL by itself learns fast at the beginning, but its final performance is capped. This is because despite collecting more samples on the environment, those come with no reward of any kind indicating what

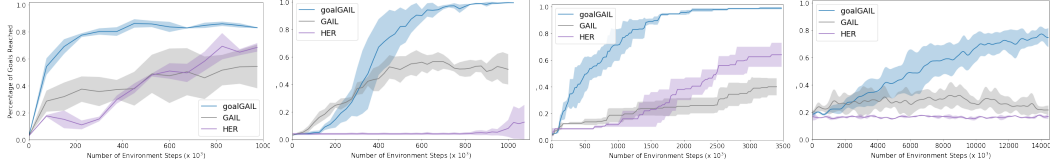

(a) Continuous Four rooms  (b) Pointmass block pusher  (c) Fetch Pick & Place  (d) Fetch Stack Two

Figure 3: In all four environments, the proposed algorithm *goalGAIL* takes off and converges faster than HER by leveraging demonstrations. It is also able to outperform the demonstrator unlike standard GAIL, the performance of which is capped.

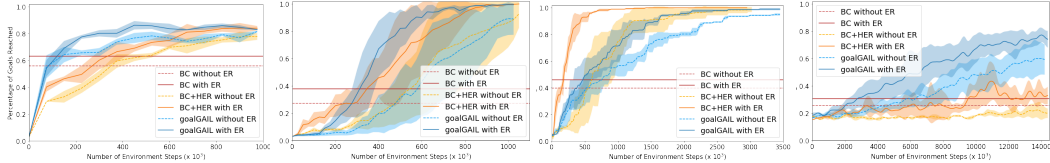

(a) Continuous Four rooms  (b) Pointmass block pusher  (c) Fetch Pick & Place  (d) Fetch Stack Two

Figure 4: Our Expert Relabeling technique boosts final performance of standard BC. It also accelerates convergence of BC+HER and *goalGAIL* on all four environments.

is the task to perform (reach the given goals). Therefore, once it has extracted all the information it can from the demonstrations it cannot keep learning and generalize to goals further from the demonstrations. This is not an issue anymore when combined with HER, as our results show.

## 5.3 Expert relabeling

Here we show that the Expert Relabeling technique introduced in Section 4.2 is an effective means of data augmentation on demonstrations. We show the effect of Expert Relabeling on three methods: standard behavioral cloning (*BC*), HER with a behavioral cloning loss (*BC+HER*) and *goalGAIL*. For *BC+HER*, the gradient of the behavior cloning loss $\mathcal{L}_{\text{BC}}$ (Equation 1) is combined with the gradient of the policy objective $\nabla_\theta \hat{J}$ (Equation 2). The resulting gradient for the policy update is:

$$\nabla_\theta \hat{J}_{\text{BC+HER}} = \nabla_\theta \hat{J} - \beta \nabla_\theta \mathcal{L}_{\text{BC}}$$

where $\beta$ is the weight of the BC loss and is annealed to enable the agent to outperform the expert.

As shown in Fig. 4, our expert relabeling technique brings considerable performance boosts for both Behavioral Cloning methods and *goalGAIL* in all four environments.

We also perform a further analysis of the benefit of the expert relabeling in the four-rooms environment because it is easy to visualize in 2D the goals the agent can reach. We see in Fig. 1 that without the expert relabeling, the agent fails to learn how to reach many intermediate states visited in demonstrations.

The performance of running pure Behavioral Cloning is plotted as a horizontal dotted line given that it does not require any additional environment steps. We observe that combining BC with HER always produces faster learning than running just HER, and it reaches higher final performance than running pure BC with only 20 demonstrations.

## 5.4 Robustness to sub-optimal expert

In the above sections we were assuming access to perfectly optimal experts. Nevertheless, in practical applications the experts might have a more erratic behavior, not always taking the best action to go towards the given goal. In this section we study how the different methods perform when a sub-optimal expert is used. To do so we collect sub-optimal demonstration trajectories by adding noise $\alpha$ to the optimal actions, and making it be $\epsilon$-greedy. Thus, the sub-optimal expert is $a = \mathbb{1}[r < \epsilon]u + \mathbb{1}[r > \epsilon](\pi^*(a|s,g) + \alpha)$, where $r \sim \text{Unif}(0,1)$, $\alpha \sim \mathcal{N}(0, \sigma_\alpha^2 I)$ and $u$ is a uniformly sampled random action in the action space.

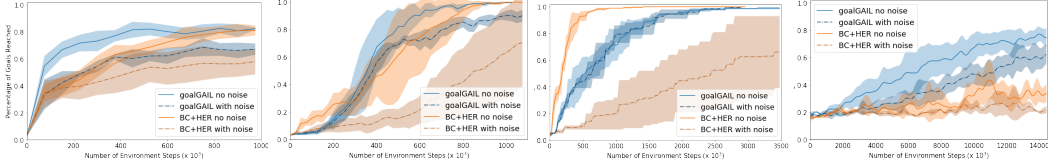

(a) Continuous Four rooms  (b) Pointmass block pusher  (c) Fetch Pick & Place  (d) Fetch Stack Two

Figure 5: Effect of sub-optimal demonstrations on *goalGAIL* and Behavorial Cloning method. We produce sub-optimal demonstrations by making the expert $\epsilon$-greedy and adding Gaussian noise to the optimal actions.

In Fig. 5 we observe that approaches that directly try to copy the action of the expert, like Behavioral Cloning, greatly suffer under a sub-optimal expert, to the point that it barely provides any improvement over performing plain Hindsight Experience Replay. On the other hand, methods based on training a discriminator between expert and current agent behavior are able to leverage much noisier experts. A possible explanation of this phenomenon is that a discriminator approach can give a positive signal as long as the transition is "in the right direction", without trying to exactly enforce a single action. Under this lens, having some noise in the expert might actually improve the performance of these adversarial approaches, as it has been observed in many generative models literature [42].

## 5.5 Using state-only demonstrations

Behavioral Cloning and standard GAIL rely on the state-action $(s, a)$ tuples coming from the expert. Nevertheless there are many cases in robotics where we have access to demonstrations of a task, but without the actions. In this section we want to emphasize that all the results obtained with our *goalGAIL* method and reported in Fig. 3 and Fig. 4 do *not* require any access to the action that the expert took. Surprisingly, in all environments but Fetch Pick & Place, despite the more restricted information *goalGAIL* has access to, it outperforms BC combined with HER. This might be due to the superior imitation learning performance of GAIL, and also to the fact that these tasks are solvable by only matching the state-distribution of the expert. We run the experiment of training GAIL only conditioned on the current state, and not the action (as also done in other non-goal-conditioned works [25]), and we observe that the discriminator learns a very well shaped reward that clearly encourages the agent to go towards the goal, as pictured in Fig. 6. See the Appendix for more details.

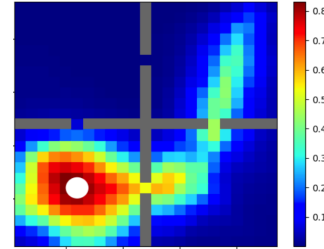

Figure 6: Output of the Discriminator $D(\cdot, g)$ for the four rooms environment. The goal is the lower left white dot, and the start is at the top right.

## 6 Conclusions and Future Work

Hindsight relabeling can be used to learn useful behaviors without any reward supervision for goal-conditioned tasks, but they are inefficient when the state-space is large or includes exploration bottlenecks. In this work we show how only a few demonstrations can be leveraged to improve the convergence speed of these methods. We introduce a novel algorithm, *goalGAIL*, that converges faster than HER and to a better final performance than a naive goal-conditioned GAIL. We also study the effect of doing expert relabeling as a type of data augmentation on the provided demonstrations, and demonstrate it improves the performance of our *goalGAIL* as well as goal-conditioned Behavioral Cloning. We emphasize that our *goalGAIL* method only needs state demonstrations, without using expert actions like other Behavioral Cloning methods. Finally, we show that *goalGAIL* is robust to sub-optimalities in the expert behavior.

All the above factors make our *goalGAIL* algorithm very suited for real-world robotics. This is a very exciting future work. In the same line, we also want to test the performance of these methods in vision-based tasks. Our preliminary experiments show that Behavioral Cloning fails completely in the low data regime in which we operate (less than 20 demonstrations).

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

# A Hyperparameters and Architectures

In the four environments used in our experiments, i.e. Four Rooms environment, Fetch Pick & Place, Pointmass block pusher and Fetch Stack Two, the task horizons are set to 300, 100, 100 and 150 respectively. The discount factors are $\gamma = 1 - \frac{1}{H}$. In all experiments, the Q function, policy and discriminator are paramaterized by fully connected neural networks with two hidden layers of size 256. DDPG is used for policy optimization and hindsight probability is set to $p = 0.8$. The initial value of the behavior cloning loss weight $\beta$ is set to $0.1$ and is annealed by $0.9$ per $250$ rollouts collected. The initial value of the discriminator reward weight $\delta_{GAIL}$ is set to $0.1$. We found empirically that there is no need to anneal $\delta_{GAIL}$ .

For experiments with sub-optimal expert in section 5.4, $\epsilon$ is set to $0.4$, $0.5$ $0.4$, $0.1$, and $\sigma_\alpha$ is set to $1.5$, $0.3$, $0.2$ and $0$ respectively for the four environments.

# B Effect of Different Input of Discriminator

We trained the discriminator in three settings:

- current state and goal: $(s, g)$
- current state, next state and goal: $(s, s', g)$
- current state, action and goal: $(s, a, g)$

We compare the three different setups in Fig. 7.

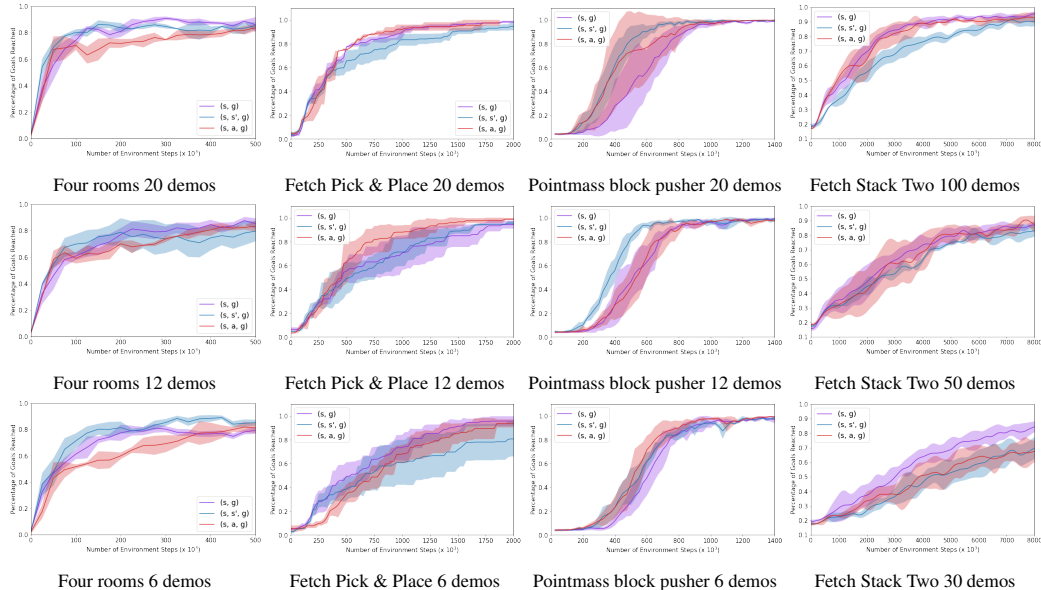

Figure 7: Study of different discriminator inputs for *goalGAIL* in four environments

