[Reviews · NeurIPS 2019]

Reviewer 1



After rebuttal comments: the authors address many questions and propose some updates to improve the quality of the paper. In my view these are minor and doable. Assuming this adjustments my overall score is increased. ________________________________________________ Originality: the paper builds on previous work and ideas used in those works (HER, BC, DDPG, GAIL). The paper argues how learning can be sped up in goal oriented sparse reward problems using demonstrations. In previous work (HER) data relabeling has been exploited and now applied for BC. Additionally, goal oriented demonstrations are combined with GAIL (goal oriented GAIL) and DDPG to learn an appropriate policy. Ultimately, the policy gradient of BC can be combined with DDPGs based on discriminator rewards in GAIL. While the paper considers different combinations of existing works it mostly convincingly shows how demonstrations can speed up learning especially in "bottlenecked" tasks. The paper does not discuss literature that purely considers demonstrations for goal oriented robot learning (on real robots) [1] Quality: the overall quality of the paper is OK, but some details remain vague. The idea of using sparse rewards depending on goal states limits the range of tasks where these kind of approaches are applicable. For example, for the FetchSlide task, or for a simple throwing task the state-dependent sparse reward will not work in my opinion. The paper also claims that scaling to real world robotic tasks of goalGAIL is potentially possible. It is difficult to see how, given the data-hungry RL setting (at least 10^6 step until convergence). While the paper convincingly shows different positive aspects of exploiting demonstrations, it is not clear how these demonstrations were collected for the simulated tasks. Clarity: the paper overall reads well. However, the techniques used are not always clearly introduced/explained, (examples: what is the objective J in the policy gradient in Sec. 4.1, also use equation numbers; say clearly what does the discriminator corresponds to when introducing GAIL, what's the main idea behind GAIL?) I'm afraid the reader requires a substantial knowledge of related literature for understanding the technical approach due to the lack of clarity while introducing the used work. Minor suggestions: use goalGAIL or gGAIL in the figures to highlight your proposed approach. Significance: the paper overall carries an interesting message, but the clarity and some vague technical details make its impact a bit hard to assess. Nevertheless, in my opinion the paper shares enough value for the research community and others may be motivated by and use the work. [1] Calinon: A Tutorial on Task-Parameterized Movement Learning and Retrieval, Intelligent Service Robotics 9 (1), 1-29, 2016

Reviewer 2



Post-response comments: I have read the response and it was informative. The new tasks are a good addition. ----------- -What do you mean with quasi-static tasks in Section 4.2. It could be a number of different things and I’m not sure I captured which one it refers to. -Using state-only trajectories instead of s-a pairs trajectories certainly seems convenient for being able to operate with demonstrations from different sources and for generalization to different agents with different properties. At the same time, I wonder if there are negative effects of doing this, as the notion of the transition function is lost. - In Figure 1, was the system starting from the same initial state for each test to each goal state? - The writing was clear and easy to follow.

Reviewer 3



Even though the proposed expect relabeling technique can augument the training data to help alleviate the reward sparsisty problem, the introduced goals, which are intermediate states, are different from the groundtruth goal and this may introduce a large number of noises especially when the true rewards are limited. I don't know how to alleviate or avoid this problem or equivalently how to guarantee that the benefit of augumenting data will be larger than the negative effect of the introduced noises. Line 171: different than->different from ------------------------------------------------------------------------------------------ Authors' response partially clarifies my concern.

Reviewer 4



# Originality This work is primarily built on Hindsight Experience Replay (HER), Behavioural Cloning (BC), and Generative Adversarial Imitation Learning (GAIL). It combined these three works by additionally proposed a new expert relabeling technique. Besides, by replacing the action by the next state they can train the algorithm with state-only demonstrations. Although the novelty of this paper is a little incremental, it combined all the stuff in a reasonable fashion. # Quality In general, the method proposed in this paper is technically and experimentally sound. On both sides, however, I still have a few questions. First, in Section 4.2, you claimed that the key motivating insight behind the idea of relabeling the expert is that ''if we have the transitions (s_t, a_t, s_{t+1}, g) in a demonstration, we can also consider the transition (s_t, a_t, s_{t+1}, g'=s_{t+k})''. Did you have a rigorously mathematical proof of this statement under which condition it is the case? Because it is well known that a globally optimal policy is not necessary to be locally optimal at each step or each sub-trajectory. Second, on the experimental part, do you have any explanation about why GAIL+HER w/wo ER is better than BC+HER w/wo ER in continuous four rooms but is worse in fetch pick&place? # Clarity The paper is generally well written. However, in order to be more reader-friendly, the authors had better reorganise the layout to keep texts and their corresponding figures/tables/algorithms on the same paper as much as possible. For example, Algorithm 1 is presented on Page 1 but its first mention is on Page 5. In addition, the paper has some minor typos: e.g., is able -> is able to (line 57); U R -> R (line 5 of Algorithm 1); form -> from (line 109); etc. # Significance The problem explored in this paper is important and the authors proposed a natural but reasonable solution to it. Built on this, I believe there are still some other directions worth exploring. For instance, as we seen in Figure 4, BC and GAIL with ER and without ER perform quite differently in the two tasks. There must be some reason behind this phenomenon. Is it task-specific or method-specific? If method-specific, what causes the difference? etc. All this should be of great interest and of assistance to the community.

[Author Response · NeurIPS 2019]

We thank the reviewers for their detailed comments. We hope this rebuttal addresses their concerns. First we clarify the problem we tackle of goal-reaching, its relevance, and properties. Then we provide a rigorous mathematical proof of the correctness of our Expert Relabeling technique. We also emphasize our other main contributions from an algorithm point of view. Finally, we include results on two considerably more complex environments, and further clarifications.

We tackle the problem of **learning a universal goal-reaching policy** [9] that, given **any goal** $g$, produces actions that lead to it. This can be specified by maximizing the indicator reward as defined in our Section 2: $r_t = 1[s_{t+1} == g]$.

- *[R1]"The idea of using sparse rewards depending on goal states"* is very extended in the literature [9, 12, 15]. As can be seen in these prior works, **many robotics problems** can be formulated as such. In particular, the FetchSlide task referred by *R1* was originally introduced in the HER paper [12], where such a sparse reward is used.

- *[R4] "the introduced goals, which are intermediate states, are different from the ground truth goals"*: in our problem statement, there are **not ground truth goals**. There is no "true reward" neither, and all our experiments only ever use the above-specified indicator reward. We are interested in learning to reach all goals equally well, and that is why our performance is evaluated in terms of the fraction of goals reached, as is common in this literature.

Here we provide a *[R5, 4] "rigorously mathematical proof of the statement"* that *"**guarantee the benefit** of augmenting data"* with **our Expert Relabeling** strategy, in the sense that it yields new $(s, a, s', g)$ tuples that could have been produced by the expert. For a discrete state-action space, with deterministic dynamics, and assuming the demonstrations are optimal, the proof reduces to a shortest-path argument in graphs:

1. By the optimality of the demonstration $(s_0, a_0, s_1, a_1, s_2, \ldots, g)$, there is no shorter path from $s_0$ to $g$.
2. By contradiction, there is no shorter path from $s_0$ to any encountered $s_t$ neither, because if such path $P' = (s_0, s'_1, \ldots s'_{t-1}, s_t)$ existed, then the path $(s_0, s'_1, \ldots, s'_{t-1}, s_t, \ldots, g)$ would be shorter than the demonstration.
3. By the same argument, there is no shorter path from $s_t$ to $s_{t+k}$ than the one that starts by going to $s_{t+1}$.
4. Therefore $(s_t, a_t, s_{t+1}, g' = s_{t+k})$ could also have been produced by the expert (the transition is optimal for $g'$).

The argument can be extended to continuous stochastic case. We will include further details in the Appendix.

On top of our study of ER, we propose a **novel algorithm, goalGAIL**, that **combines and outperforms both HER and GAIL**. We also show that the algorithm is **robust to sub-optimal demonstrations** and that it can also **leverage state-only demonstrations**, which are very practical in robotics. Note that as long as the discriminator receives the state and next state $(s, s', g)$ as input there is no concern that *[R2]"the notion of transition might be lost"* because this tuple captures the kind of transitions that the expert performs towards the goal $g$. As can be seen in Fig. 8 of our submitted Appendix, there are **no negative effects** on the studied tasks. BC + HER is not suited for these situations, and therefore the comparative performance of goalGAIL and BC + HER is of limited interest. We agree with R5 that our results should spark further research directions in the community about BC v.s. GAIL.

*[R2] "Performing experiments with more complex domains"*: we added two more complex tasks: BlockPusher and Stack2. In BlockPusher a point-mass not only navigates itself, but also displaces a Block. In Stack2 a Fetch robot stacks two blocks on a desired spot, as done in [35]. These results bolster the conclusions of our paper. Furthermore *[R1] "scaling to real robot scenarios"* is not too far if we consider that 1M steps corresponds to 6h of real robot time [35].

**Figure 1:** Experimental results on BlockPusher(row 1) and Stack2(row2). Column 2, 3, 4 correspond to the study in Fig. 3, 4 and 6 respectively in the submitted paper.

- *[R1] "What is the objective J"*: it's the expected cumulative reward. We will link this to line 20 of our algorithm.
- *[R1]* We will include a paragraph on GAIL in the background section to make the paper more self-contained.
- *[R1]* The work on "Task-parameterized movement learning" is very interesting, and we will explore this literature.
- *[R2]* **Quasi-static tasks** can be performed arbitrarily slow. This is the case for most robotics manipulation. If we also care about velocities, we can still use our framework by including velocities in the goal space.
- *[R2]* The system does **not need to start always from the same state**. This was the case only for the four-room experiment. In fetch robot experiments, the block positions are uniformly sampled at every rollout.

[Meta-Review · NeurIPS 2019]

This paper studies imitation learning in the goal-conditioned settings and the proposed method leverages previous ones in a sensible way. The studied problem is of interest to the community, and the reported results are encouraging.